# Navigating Gender-Affirming Healthcare in Adverse Political Climates: Experience from a University-Based Transgender Clinic in Türkiye

**DOI:** 10.3390/healthcare13202591

**Published:** 2025-10-14

**Authors:** Aslıhan Polat, Hanife Yılmaz Abaylı, İlay Dalkıran, Aila Gareayaghi, Seray Karakoç, Nezihe Gül, Zeynep Büyükkaraca, Ezgi Şişman, Seher Kocaayan, Mehtap Güngör, Berrin Çetinarslan, Zeynep Cantürk, Alev Selek, Emre Gezer, Mehmet Sözen, Mustafa Melih Çulha, Naci Burak Çınar, Emrah Yaşar, Murat Şahin Alagöz, Şener Gezer, Özge Senem Yücel Çiçek, Seher Şirin

**Affiliations:** 1Department of Psychiatry, Faculty of Medicine, Kocaeli University, 41001 Izmit, Türkiye; ilay.dalkiran@gmail.com (İ.D.); aila.gareayaghi@kocaeli.edu.tr (A.G.); s.seraykarakoc@gmail.com (S.K.); nezihe.gul@kocaeli.edu.tr (N.G.); seherkocaayan@gmail.com (S.K.); mehtapgungorcaglar@hotmail.com (M.G.); 2Department of Psychiatry, Şemdinli State Hospital, 30800 Hakkari, Türkiye; h.hanifeyilmaz@gmail.com; 3Department of Psychiatry, Çorlu State Hospital, 59850 Tekirdağ, Türkiye; zeynepbkaraca@gmail.com; 4Department of Psychiatry, Kocaeli City Hospital, 41060 Izmit, Türkiye; drezgisisman@gmail.com; 5Department of Endocrinology, Faculty of Medicine, Kocaeli University, 41001 Izmit, Türkiye; barslan@kocaeli.edu.tr (B.Ç.); zeynepcanturk@hotmail.com (Z.C.); alev.selek@kocaeli.edu.tr (A.S.); emre.gezer@kocaeli.edu.tr (E.G.); mehmet.sozen@kocaeli.edu.tr (M.S.); 6Department of Urology, Faculty of Medicine, Kocaeli University, 41001 Izmit, Türkiye; melih.culha@kocaeli.edu.tr (M.M.Ç.); burak.cinar@kocaeli.edu.tr (N.B.Ç.); 7Department of Plastic, Reconstructive and Aesthetic Surgery, Faculty of Medicine, Kocaeli University, 41001 Izmit, Türkiye; emrah.yasar@kocaeli.edu.tr (E.Y.); sahinalagoz@yahoo.com (M.Ş.A.); 8Department of Gynecology, Faculty of Medicine, Kocaeli University, 41001 Izmit, Türkiye; sener.gezer@kocaeli.edu.tr (Ş.G.); ozge.yucelcicek@kocaeli.edu.tr (Ö.S.Y.Ç.); 9Department of Otorhinolaryngology, Faculty of Medicine, Kocaeli University, 41001 Izmit, Türkiye; ugurseher@hotmail.com

**Keywords:** gender-affirming care, gender dysphoria, multidisciplinary team, transgender health, Türkiye

## Abstract

Background/Objectives: Gender-affirming care is a complex, multidisciplinary process that has gained increasing recognition worldwide. This practice report presents the unique clinical model developed at the Kocaeli University Hospital Gender Dysphoria Clinic, one of the pioneering centers in Türkiye. Methods: An experience-based descriptive approach was adopted to capture two decades of clinical experience, institutional processes, and socio-political challenges not fully reflected in systematic reviews. The article reflects on the authors’ direct practice in a university-based transgender health clinic in Türkiye, structured around the multidisciplinary team model, patient pathways, institutional processes, and the sociopolitical context of care. Results: The Kocaeli University model integrates psychiatry, endocrinology, surgery, nursing, social work, and legal consultation. Since 2004, the clinic has evaluated and treated hundreds of transgender individuals and produced numerous theses and peer-reviewed publications. Key strengths include a structured board system, the training of psychiatry residents, and close interdepartmental collaboration. Conclusions: This model illustrates how gender-affirming care can be effectively implemented in a challenging sociopolitical context. Sharing this experience may inform global practices and support the dissemination of multidisciplinary approaches to transgender health.

## 1. Introduction

Healthcare approaches to individuals experiencing gender incongruence have undergone significant transformation over the years. In the 1960s and 1970s, the first gender clinics were established in Europe and the United States, and initial treatment protocols for gender affirmation treatments began to emerge [1]. Today, the term “gender-affirming care” refers to a comprehensive model of medical, psychological, and social services that affirm and support the identities of transgender and gender-diverse (TGD) individuals [2].

This model of care has been shaped by decades of clinical experience and research and is currently guided by up-to-date international standards. These include the World Professional Association for Transgender Health’s (WPATH) Standards of Care Version 8 (SOC-8) published in 2022, and the European Network for the Investigation of Gender Incongruence (ENIGI) protocol, which emphasizes individualized, flexible, and multidisciplinary care involving psychiatry, endocrinology, and surgery [3,4].

Globally, gender-affirming care models increasingly emphasize multidisciplinary collaboration, patient-centered approaches, and rights-based frameworks. Comparative studies show that while high-income countries often benefit from specialized centers and strong legal protections, access remains highly uneven in middle- and low-income settings. In many countries, continuous psychiatric follow-up is not mandated, yet its importance is increasingly recognized, particularly in addressing comorbidities and psychosocial challenges. Therefore, adapting international standards such as WPATH SOC-8 and ENIGI to local socio-political contexts is critical for ensuring inclusive and sustainable care.

Across the globe—including in countries with well-established human rights protections—healthcare for transgender individuals is often discussed not only within medical circles but also in social and political arenas. Transgender individuals in Türkiye (Turkey), a secular yet predominantly Muslim country, have long accessed gender-affirming healthcare services, yet these services have recently become increasingly difficult to deliver due to growing political and social barriers [5]. While some hospitals still offer psychiatric or surgical support to transgender patients, in today’s Türkiye, being a transgender person is difficult—but so is being a physician who serves them [6,7].

Established in 2004, Kocaeli University Gender Identity Clinic (KoUGIC) is one of the leading multidisciplinary centers for gender-affirming care in Türkiye and serves as a major national referral hub with over 200 patients being followed up. KoUGIC serves not only as a healthcare provider but also as an academic center engaged in education and research [8]. This article aims to offer both a clinical reflection and a critical commentary on the sociopolitical challenges surrounding gender-affirming healthcare in Türkiye. Our goal is to contribute to the development of evidence-based health policies while also presenting a replicable clinical model for other centers and professionals.

## 2. Gender-Affirming Care in a Politicized Context: Challenges and Resilience in Türkiye

Türkiye’s societal stance on gender diversity reflects a cultural ambivalence—caught between Eastern conservatism and Western liberalism. For example, the case of a widely beloved public figure who initially faced performance bans due to their gender identity but later gained mainstream popularity illustrates the inconsistent and contradictory public attitudes [9].

Legal regulations on gender transition in Türkiye date back to 1988, when Article 29 of the former Civil Code (No. 743) allowed legal gender to change with a medical board report, making Türkiye the fifth European country to recognize transition. The current Civil Code (No. 4721, Art. 40, 2002) added stricter criteria, requiring medical approval that transition was necessary for mental health and that the individual was permanently infertile. In 2017, the infertility requirement was abolished, marking a shift toward a more rights-based framework [10]. Although legal and normative advances have been achieved, Türkiye has shifted toward a more conservative sociopolitical stance, with legislative discourse reflecting traditionalist Islamic values. In the 2024 Rainbow Europe Index, Türkiye ranked third lowest among 48 countries, scoring poorly on all LGBTI rights indicators [11]. Official policy frames terms such as “gender,” “gender identity,” “LGBT,” “SOGIESC,” and “comprehensive sexuality education” as threats to the family and society, excluding them from national and international documents, training, and awareness programs. This stance emphasizes protecting the biological characteristics of the two sexes and avoiding narratives that normalize diverse gender identities.

Despite low public visibility, the LGBTQ community in Türkiye has increasingly been targeted by negative political discourse and institutional rhetoric. Since 2015, Pride parades have been banned, while state-backed rallies against LGBTQ rights have gained ground. In 2023, the Minister of Family and Social Services called for a “struggle against LGBT rights” [12]. Most recently, national campaigns have framed LGBTQ identities as threats to the traditional family, with one initiative declaring the coming year the “Year of the Family” and describing LGBTQ issues as “one of the most serious threats to the existence of the family” [13].

Türkiye may be described as a society shaped by competing forces of traditionalism and modernity, with tensions between these orientations intensifying over the past decade. Within this context, the LGBTQ population remains at the margins of a continuum between a conservative majority and a secular minority. Historically, nonconformity to dominant norms of masculinity has been met with social repression, a trend now amplified in political discourse. Unlike progress seen at the EU and UN levels, Türkiye still lacks comprehensive anti-discrimination laws addressing sexual orientation and gender identity [14].

## 3. The KoUGIC Model

Founded in 2004, the clinic began with psychiatry and endocrinology, later expanding to include urology, which performed the university’s first gender-affirming surgery in 2008. In subsequent years, plastic surgery, gynecology, and otorhinolaryngology were integrated into the model, reflecting the need for comprehensive, multidisciplinary care. The establishment of a structured Gender Affirmation Board was not only designed to ensure consensus-based decision-making, but also to distribute responsibility among different specialties and safeguard clinicians in a restrictive socio-political climate. The core decision-making board subsequently grew to involve plastic surgery, obstetrics and gynecology, and otorhinolaryngology (notably for voice and speech interventions). Forensic medicine briefly took part but was later excluded, as the board’s scope centers on medical rather than legal assessments [8].

KoUGIC provides care for individuals over the age of 18. The process begins at a dedicated gender identity outpatient clinic operating on weekdays. After an initial psychiatric evaluation, applicants undergo detailed psychometric testing by a clinical psychologist. They are then referred to endocrinology and, as appropriate, to urology or gynecology. All consultations are coordinated by an experienced psychiatric nurse serving as case manager. For those diagnosed with gender dysphoria and accepted into the program, individualized follow-up plans are developed to guide the transition process.

Individuals in the program must complete at least one year of real-life experience, integrated with psychiatric follow-up through group or individual psychotherapy, before being eligible for gender-affirming hormone treatment under the Gender Affirmation Board (GAB) supervision. After GAB approval, a minimum of one year of hormone therapy with regular endocrinology visits and ongoing psychiatric sessions is required before the board’s final evaluation for gender-affirming surgery. Psychiatric follow-up remains mandatory throughout, with access to group therapy, speech therapy, and educational sessions. Once GAB grants final approval, individuals must obtain court authorization for surgical procedures, which are covered by public insurance. The clinic accepts applicants at all stages—first-time visitors, those on hormones, post-surgical individuals, and court referrals. As a public university hospital, KoUGIC provides most services under Türkiye’s national health insurance system, including psychiatric assessments, hormone therapy, and surgical interventions. This framework ensures access for patients with limited financial resources. Nevertheless, indirect costs—such as travel, accommodation, or medications not fully reimbursed—remain barriers, especially for those from rural or socioeconomically disadvantaged areas. For first-time applicants, the full transition process typically takes around two years, though duration varies by individual needs.

In addition to binary transgender individuals, the situation of non-binary people in Türkiye deserves attention. Due to pervasive stigma and discrimination, many non-binary individuals avoid seeking healthcare even for daily medical needs. Those who present to hospitals usually request gender-affirming interventions such as hormones or surgery, which are less relevant for non-binary people. Being non-binary in Türkiye is often even more difficult than being binary transgender, as social recognition and acceptance are limited. Some non-binary individuals initially feel compelled to pursue binary-oriented treatments due to societal pressures. However, during group psychotherapy sessions at KoUGIC, several participants have recognized and embraced their non-binary identity. In such cases, they typically withdraw from hormone or surgical requests while continuing with individual psychotherapy when support is needed.

In Türkiye, gender-affirming hormone therapy may only be initiated by an endocrinologist following psychiatric approval and a formal board decision. A recent legal amendment stipulates that transgender hormone treatment can be prescribed only to individuals over the age of 21 [15]. Gender reassignment surgery requires both a medical board report and court authorization (Turkish Civil Code No. 4721, 22 November 2001, Article 40; Official Gazette No. 24607). Most recently, a government regulation has centralized decision-making: internal institutional boards are no longer recognized, and gender reassignment approvals can only be granted by the state-designated Medical Indications Assessment Board for Gender Reassignment.

At KoUGIC, our multidisciplinary board continues to evaluate and prepare patients; however, the report we produce is considered a medical opinion rather than an authorization. The final legal decision for gender-affirming surgeries rests with the court. In practice, this means that even if our board deems a patient suitable for surgery, the court may deny permission. This dual structure limits clinical autonomy and can create delays or inconsistencies between medical expertise and legal authority.

At KoUGIC, (GAB oversees both hormonal and surgical transition decisions through a multidisciplinary and consensus-based process. If unanimity cannot be achieved, cases are deferred for re-evaluation, ensuring peer support and shared responsibility among clinicians. With informed consent, medical students may observe board meetings, foster education and reducing stigma. The wider multidisciplinary team also includes psychiatric nurses, social workers, rehabilitation specialists, and infectious disease and gastroenterology experts. Social workers assist patients with legal, occupational, and social service access, while rehabilitation specialists support postoperative recovery. Infectious disease collaboration provides patient education and monitoring for sexually transmitted and procedure-related infections. Follow-up extends beyond medical eligibility to address psychological resilience, overall health, and social support. Group psychotherapy is prioritized, with individual therapy offered when group participation is not feasible [16].

The KoUGIC integrates telepsychiatry as an important complementary component of its gender-affirming care model. An online outpatient clinic, initially established during the COVID-19 pandemic, remains actively operational and serves a wide range of patients who may face barriers to attending in-person appointments [17]. The Gender Identity Clinic receives patients from across Türkiye seeking comprehensive gender-affirming care. To address these needs, psychiatric residents and nurses provide online follow-up and psychosocial support in addition to in-hospital services. This hybrid model ensures continuity of care, especially for those traveling long distances, while telepsychiatry expands access to underserved regions. Recent evidence also shows that online group psychotherapy reduces shame and fosters self-acceptance among transgender individuals facing access barriers [18].

This multidisciplinary approach model not only supports individuals navigating gender-affirming care but also empowers healthcare professionals by fostering a collegial, supportive, and collaborative environment. Interdisciplinary teamwork and shared decision-making enhance care quality and contribute to a sustainable and inclusive model of healthcare delivery [19].

As an academic center committed to education and research, KoUGIC has contributed multiple peer-reviewed articles, numerous unpublished dissertations and theses, and authored the first and only Turkish textbook on the subject, “Multidisciplinary Approach to Gender Dysphoria” [20]. The research and publications of KoUGIC are presented in Table 1.

Although international guidelines do not mandate family involvement in adult gender-affirming care, KoUGIC adapts to Türkiye’s socio-cultural context by engaging families through regular information and support meetings. These sessions dispel misinformation, foster dialog, and enable relatives to provide practical and emotional support during transition. Individual consultations and psychiatric support are also available for close contacts when needed. In parallel, KoUGIC ensures that patients and caregivers receive accurate, timely guidance on surgical options, complications, and post-operative care through regular multidisciplinary meetings led by psychiatry with input from endocrinology, urology, gynecology, plastic surgery, and ENT specialists.

## 4. Individual Psychotherapy as a Component of Gender-Affirming Care

Although not required by WPATH SOC-8, individual psychotherapy remains valuable in gender-affirming care. At KoUGIC, psychiatrists provide biopsychosocial support addressing both gender identity and common comorbidities (e.g., depression, anxiety, eating disorders, substance use). Psychotherapy reduces dysphoria, especially where stigma and lack of support heighten distress and maladaptive coping [34]. Current evidence underscores the clinical need for individual interventions, with studies such as Liu et al. (2023) showing that greater gender dysphoria is strongly linked to higher levels of depression, anxiety, suicidal ideation, and suicide attempts [35].

At the clinic, the real-life experience (RLE) phase serves as a key psychotherapeutic tool, preparing individuals for hormone therapy and surgeries while allowing clinicians to assess their social, psychological, and functional adaptation alongside the adequacy of personal, familial, and institutional support systems [36,37]. At KoUGIC, gender-affirming care integrates inpatient and outpatient support: the 21-bed psychiatry unit ensures privacy, affirming communication, and hormone continuity, while individual psychotherapy offers diagnostic clarification, transition guidance, coping support, family education, and academic/occupational counseling. For those not ready for group therapy, it serves as the main psychosocial intervention [38].

## 5. Group Psychotherapy as a Component of Gender-Affirming Care

KoUGIC considers group psychotherapy an integral component of the gender-affirmation process. Conducted monthly in an open self-help group format, these sessions cover a broad range of topics, including gender identity and sexual orientation, family and social relationships, stages of the transition process, medical follow-ups, legal procedures, personal safety, problem-solving skills, increasing resilience through improving coping mechanisms, etc. [16,39].

At KoUGIC, group psychotherapy brings together individuals at different stages of transition, fostering peer support and shared learning. Sessions provide a safe space where chosen names and pronouns are respected, reinforcing gender identity. Trans-affirmative practice emphasizes inclusivity, safety, and respect while rejecting coercive or conversion-based approaches. By addressing power dynamics, discrimination, and diversity of identity within the group, therapists create a supportive environment that strengthens psychosocial adjustment [40].

At KoUGIC, group psychotherapy offers a safe, affirming space where gender-affirming language and pronouns are consistently used. Therapists address both internalized and structural transphobia, while peer support encourages authentic self-expression and resilience. Through shared experiences, participants gain clarity about transition stages and strengthen psychological well-being [41]. KoUGIC’s group model supports participants, improves diagnostic accuracy before medical interventions, and—through peer support—reduces isolation, builds resilience, and counters transphobia, fostering long-term psychosocial well-being.

## 6. Nursing Care in Gender-Affirming Healthcare

International guidelines often overlook detailed nursing roles in gender-affirming care, yet numerous studies emphasize their vital contributions to ensuring continuity and quality of medical and psychosocial support during transition [42]. As the American Nurses Association emphasizes, nurses caring for gender-diverse individuals go beyond technical tasks, providing emotional support, education, and guidance throughout the transition process [43]. As the professionals with the most sustained patient contact, nurses play a vital role in gender-affirming care by receiving training, adopting affirming approaches, and building trust with transgender individuals. At Kocaeli University, psychiatric nurses serve as co-therapists in group psychotherapy, provide telehealth counseling, and engage families to strengthen social support, thereby bridging clinical care with real-life experiences and enhancing continuity and accessibility of services.

## 7. Family Support Groups in Gender-Affirming Care

In Türkiye, as in other collectivist societies, family remains a central source of psychosocial support into adulthood, and evidence shows that affirmative family involvement is a key protective factor influencing mental health and continued engagement with care [44]. Contemporary research on transgender mental health consistently highlights family support as the strongest predictor of reduced symptoms of depression and anxiety [45].

International guidelines provide little direction on family involvement in adult gender-affirming care, yet in Türkiye’s socio-cultural context, family remains central, as many adults continue to live with their relatives. At the Kocaeli University Gender Identity Clinic, this reality is addressed through regular family information and support meetings, as well as individualized consultations, which aim to educate families, dispel misinformation, and foster supportive roles during medical treatments and post-surgical recovery. A recent local psychiatry thesis confirmed family support as the single most important determinant of quality of life among transgender individuals in Türkiye. Clinical cases vividly illustrate the dual nature of familial involvement: while one trans woman patient was tragically murdered by her family due to rejection, another trans man’s family organized a mevlit—a traditional Islamic prayer ceremony—to celebrate and affirm his male identity. These examples underscore both the profound risks and the protective potential of family involvement in the Turkish context.

## 8. Navigating Education and Employment: Societal Barriers and Clinical Support

In Türkiye, trans individuals encounter structural barriers in education and employment. Stigma, bullying, and gendered rules (e.g., restroom access, uniforms) disrupt schooling and contribute to dropout. Discrepancies between legal documents and affirmed gender often create fears of forced disclosure, limiting job applications and career prospects. In the workplace, transphobia—ranging from hiring discrimination to harassment—undermines both self-expression and long-term employment [46].

Conducting transition and assessment processes in a university setting enriches medical education by familiarizing students with the psychological, physical, and social health needs of transgender individuals. This exposure not only sharpens clinical skills but also fosters empathy, ethical awareness, and sensitivity to gender diversity. Students thus acquire both technical competence and a human rights perspective, shaping socially responsible medical practice. Yet, the education of healthcare providers in this area remains insufficient: medical curricula often lack comprehensive LGBT+ content, and a U.S. study reported that 65% of medical students considered their institution’s LGBT+ training inadequate, with even less emphasis on transgender health [47]. Individuals with gender dysphoria face stigma, discrimination, and minority stress that heighten mental health risks, which are further compounded by inadequate care and disrespectful clinical interactions such as misnaming, inappropriate language, or trivialization of concerns [47,48]. In contrast, evidence shows that targeted, structured training for health professionals can reduce stigmatizing attitudes and improve the quality of care provided [49,50]. At our university, psychiatry residents complete a four-month rotation in the Gender Identity Clinic, gaining hands-on experience in assessment, care pathways, and medical board reporting. With informed consent, patients may encounter observing physicians or research participation, and the clinic has produced numerous theses and publications that contribute to the academic literature.

## 9. Expanding Knowledge and Practice in Multidisciplinary Gender-Affirming Care

Accurate and timely information on surgical options, risks, and postoperative care is essential for informed decision-making, adherence, and satisfaction, yet many transgender individuals lack reliable medical guidance and instead turn to potentially inaccurate online sources [51]. Structured, clinician-led education is crucial, as reliable preoperative information enhances psychological readiness, fosters realistic expectations, and improves coping with dysphoria, as shown by Poceta et al. (2019) [52]. At our institution, multidisciplinary information sessions led by psychiatry with endocrinology, plastic surgery, and ENT specialists provide patients and caregivers with reliable guidance on hormone use, surgical options, potential complications, and recovery.

Our clinic’s multidisciplinary council holds regular case discussions and provides ongoing training on gender-affirming care, language, and sensitivity for healthcare staff and students. While some universities in Türkiye have restricted mention of “gender identity and sexual orientation” in the Hippocratic Oath, many students continue to recite the full version, affirming their commitment to equality and visibility [53].

These integrated efforts equip healthcare providers and trainees to deliver safe, respectful, and inclusive care, while upholding the rights of individuals with gender incongruence and contributing to the reduction in stigma and discrimination in society.

## 10. Each Department’s Approach to Gender-Affirming Care

The routine laboratory tests, examined conditions, and gender-affirming practices of each department are summarized in Table 2.

## 11. Gender-Affirming Care Beyond ENIGI: The KoUGIC Model in the Turkish Sociocultural and Legal Landscape

While the ENIGI protocol provides a comprehensive, individualized framework for gender-affirming care, its full implementation in Türkiye is constrained by legal, societal, and institutional barriers [4]. The most significant limitation is the legal requirement for permanent infertility to obtain gender recognition, directly conflicting with ENIGI’s rights-based approach. Fertility preservation is prohibited, preventing KoUGIC from offering such services, though psychosocial counseling and information are provided in anticipation of future legal reforms. Multidisciplinary decision-making, another ENIGI hallmark, may cause delays in Türkiye; KoUGIC mitigates this by streamlining services within one clinic to reduce waiting times.

Beyond legal and procedural hurdles, stigma and discrimination remain pervasive [54]. Whereas ENIGI emphasizes an affirming environment, many transgender individuals in Türkiye face prejudice that undermines healthcare access and psychological well-being. KoUGIC addresses this through family counseling and support groups, which help reduce isolation and encourage family engagement despite cultural challenges. Additional support, such as explanatory letters to employers, also helps mitigate systemic barriers in education and employment, enabling patients to pursue desired careers. Importantly, comprehensive care is associated with improved surgical outcomes, while addressing co-occurring mental health and substance use issues pre- and post-surgery remains essential [55].

Our experience also underscores the urgent need for enhanced provider education in transgender health. Globally and locally, lack of knowledge and stigmatizing language persist in medical settings, highlighting a critical gap [44,47]. The multidisciplinary structure of KoUGIC fosters direct engagement, building staff and student competence while reducing stigmatization. Symbolically, advocating for the explicit inclusion of “gender identity and sexual orientation” in the Hippocratic Oath further reflects our clinic’s commitment to systemic change.

## 12. Discussion

This article is intentionally descriptive in nature, presenting nearly two decades of institutional and clinical experience at KoUGIC rather than new statistical analyses. While the manuscript does not provide systematic outcome data, such findings are being reported in separate studies. Transgender individuals face significant challenges beyond gender incongruence itself, including stigma, discrimination, social exclusion, and marginalization, which limit access to education, employment, healthcare, legal rights, and family or partner support [56,57]. In some cases, transphobia escalates to violence, directly endangering physical safety [56]. These stressors contribute to markedly higher rates of depression and anxiety among transgender individuals [58,59], whereas access to gender-affirming care, including hormone therapy and surgery, substantially reduces such mental health burdens [60]. The WPATH Standards of Care Version 8 (SOC-8) emphasizes that psychiatric or psychological services should be accessible and supportive but not obligatory for all [3]. Yet in Türkiye, where stigma, discrimination, and violence are pervasive, psychiatric evaluation and support hold additional importance [44,61].

Within this legal framework, psychiatric reports remain a prerequisite for initiating hormone therapy and surgeries, which often leads individuals to engage with psychiatry mainly for procedural approval rather than genuine psychosocial care. At KoUGIC, however, the routine integration of psychotherapy transforms this requirement into a truly supportive model of care. Despite the lack of standardized national protocols and the constraints of a restrictive sociopolitical climate, KoUGIC has emerged as a key referral center, providing comprehensive, multidisciplinary, and gender-affirming healthcare.

Our findings mirror global evidence linking minority stress to adverse mental health outcomes, including depression, anxiety, and suicidality [58,59,62], while underscoring that access to affirming care alleviates these difficulties [60]. KoUGIC’s psychiatric assessment protocols extend beyond gatekeeping, incorporating psychometric and projective testing to address holistic psychosocial needs. By separating evaluative and therapeutic functions across different clinicians, the clinic actively mitigates the perception of psychiatry as a barrier to transition. Individual psychotherapy, grounded in a biopsychosocial approach, not only supports gender identity exploration but also manages comorbid conditions such as depression, anxiety, and substance use, frequently intensified by minority stress [35,63]. Structured emphasis on the Real-Life Experience (RLE) phase further ensures social and psychological readiness before medical interventions [64,65].

Group psychotherapy constitutes another central pillar, offering peer support and a culturally sensitive space for authentic gender expression and social integration [39].

Nursing care provides continuity by bridging institutional services and daily life, ensuring adherence and psychosocial support [66]. The central role of family support in Türkiye’s collectivist culture is also directly addressed. Through structured family meetings and individualized consultations, the clinic engages relatives to strengthen psychosocial networks, with family affirmation shown to be a critical protective factor for mental health outcomes [44,45]. Such engagement directly enhances the feasibility and success of medical and surgical interventions [67,68].

In sum, KoUGIC illustrates the resilience and adaptability required to sustain comprehensive gender-affirming healthcare in restrictive sociopolitical contexts. By integrating psychiatric, psychotherapeutic, nursing, and family-centered interventions, the clinic not only supports individual well-being but also contributes to broader efforts toward reducing stigma and advancing social acceptance.

### 12.1. Limitations

Gender-affirming care in Türkiye remains constrained by restrictive laws, including the ban on fertility preservation [69] and the Civil Code’s sterilization requirement for legal gender recognition, as well as court and protocol burdens that delay care. Systemic challenges include a shortage of trained specialists, long wait times, inconsistent insurance coverage and geographic inequities [70]. Societal stigma further undermines access, with most families initially unsupportive, no legal protections against discrimination, and widespread provider bias and knowledge gaps [71,72]. Together, these barriers limit the reach of KoUGIC and highlight the urgent need for policy reform, capacity building, and public education. Another challenge is the limited training opportunities for healthcare providers. Medical curricula in Türkiye often lack comprehensive content on transgender health, and stigma among providers may compromise the quality of care. These factors highlight systemic barriers in practice and underline the need for enhanced professional education. For surgical and hormonal procedures, this article reports only the overall number of cases. Although follow-up data on outcomes, complications, and patient satisfaction are available within our center, these findings will be presented in separate focused studies. This represents a limitation of the current article, as it was designed primarily as a descriptive practice report, but future publications will provide these outcome measures in detail.

### 12.2. Future Directions

The future of gender-affirming care in Türkiye requires reforms in fertility preservation, surgical innovation, and professional education. Legal changes enabling sperm, oocyte, and embryo cryopreservation would protect reproductive autonomy [69,73]. Adoption of minimally invasive and robotic techniques, alongside collaboration with international centers, could improve surgical outcomes and expand options, including advanced procedures and experimental transplantation [74]. Equally vital is closing knowledge gaps in healthcare training, as studies show high levels of bias and insufficient LGBT health coverage among educators [72]. Integrating transgender health into curricula, residency, and continuing education, while promoting inclusive, team-based models like KoUGIC, will strengthen access, quality, and sustainability of care.

## 13. Conclusions

The KoUGIC exemplifies a resilient model of gender-affirming healthcare in Türkiye’s restrictive sociopolitical context, showing that inclusive, high-quality care can thrive despite systemic barriers. With its multidisciplinary structure—psychiatry, endocrinology, surgery, nursing, psychology, and social work—it addresses both medical and psychosocial needs while actively countering stigma and discrimination. Family counseling, central in Türkiye’s collectivist culture, fosters acceptance and protects psychological well-being, while rigorous psychiatric, psychometric, neurocognitive, and biological assessments ensure diagnostic accuracy and readiness for interventions. Integrated psychotherapy reduces perceptions of psychiatry as gatekeeping and instead provides safe, affirming spaces that counter the harms of conversion efforts, which are linked to depression, suicidality, and substance use [35,64,65]. Early identification of psychiatric comorbidities and targeted interventions for substance use, especially smoking, are emphasized as crucial for surgical and long-term outcomes [66,67]. Additional components, including structured voice therapy referrals and neurobiological evaluations (e.g., MRI, digit ratio), further highlight the integration of psychosocial and biological dimensions within this comprehensive model.

To conclude, this work reflects a multidisciplinary effort sustained under challenging conditions, representing a unique model in Türkiye and one of the few worldwide. Beyond providing care for transgender individuals, it also fosters knowledge exchange and solidarity among healthcare professionals engaged in this field. By sharing our experience, we aim to highlight the importance of multidisciplinary collaboration in ensuring not only high-quality patient care but also professional growth and collective resilience.

## Figures and Tables

**Table 1 healthcare-13-02591-t001:** KoUGIC Research and Publications.

Author(s)	Title	Year	Type	Journal/Publisher
Polat, A., Ağaoğlu, F. [8]	Gender dysphoria: Kocaeli University experience	2019	Article	Alpha Psychiatry, 20(1), 101–109. doi:10.5455/apd.299184
Polat, A. (Ed.) [20]	Gender Dysphoria: Principles of Multidisciplinary Approach (in Turkish)	2020	Book	Nobel Tıp Kitabevleri, Ankara
Gkiouler et al. [21]	Predictors of Family Attitude of the Individuals with Gender Dysphoria	2019	MD thesis	Kocaeli University School of Medicine, Psychiatry Department
Sirin, S., Polat, A. [22]	The association between subjective and objective masculine vocal quality in hormone-naïve trans male individuals	2019	Article	ENT Updates, 9(3), 219–226
Alioğlu, F. [23]	Descriptive Psychiatric Evaluation of the Individuals with Gender Dysphoria	2020	MD thesis	Kocaeli University School of Medicine, Psychiatry Department
Sirin, S., Polat, A. [24]	Voice-related outcomes after long-term androgen treatment in trans males	2020	Article	Journal of Health Sciences of Kocaeli University, 6(1), 53–58
Sirin, S., Polat, A., Alioğlu, F. [25]	Psychometric evaluation of adapted Transsexual Voice Questionnaire for Turkish trans male individuals	2020	Article	Journal of Voice, doi:10.1016/j.voice.2020.01.023
Gezer, E., Piro, B., Cantürk, Z., Çetinarslan, B., Sözen, M., Selek, A., Polat, A., Seal, L.J. [26]	The comparison of gender dysphoria, body image satisfaction, and quality of life between treatment-naive transgender males with and without polycystic ovary syndrome	2021	Article	Transgender Health (Advance online publication)
Sirin, S., Polat, A., Alioğlu, F. [27]	Voice-related gender dysphoria: quality of life in hormone-naive trans male individuals	2021	Article	Alpha Psychiatry, 21(1), 53–60. doi:10.5455/apd.41947
Şener, J. [28]	Psychiatric Comorbidities in Transgender Individuals Followed in Kocaeli University Hospital’s Gender Dysphoria Unit	2022	MD thesis	Kocaeli University School of Medicine, Psychiatry Department
Kara, C. [29]	The Relationship Between Existential Anger and Life Projects in Trans Persons	2022	MSc thesis in nursing	Istanbul Okan University Graduate Education Institute, Department of Nursing
Yılmaz Abaylı, H. [30]	Neuropsychological Profile of Transgender Individuals Applying to the Gender Dysphoria Unit of Kocaeli University Hospital	2024	MD thesis	Kocaeli University School of Medicine, Psychiatry Department
Şişman, E., Güngör. M., Gareayaghi, A., Yılmaz Abaylı, H., Polat, A. [31]	Predictors of transphobia and attitudes toward transgender individuals among nurses in Türkiye: a cross-sectional study	2025	Article	Healthcare (MDPI, Basel, Switzerland), doi: 10.3390/healthcare13121474
Karakoç, S. [32]	The Role of Voice and Body Image in Predicting Quality of Life and Psychological Well-Being among Transgender Men	2025	MD thesis	Kocaeli University School of Medicine, Psychiatry Department
Polat et al. [33]	Family attitude toward transgender people in Turkey: experience from a secular Islamic country	2005	Article	International Journal of Psychiatry in Medicine, 35, 383–393

**Table 2 healthcare-13-02591-t002:** Each department’s approach to gender-affirming care.

Department	Routine Laboratory Tests	Examined Conditions	Gender-Affirming Practices	Number of Cases
Endocrinology	Hormone profile: Testosterone, estradiol, LH/FSH, and other relevant hormones. Routine biochemistry: Liver and kidney function, blood glucose, etc. Vitamin D level Prolactin level Lipid profile	Thrombosis risk factors: Before hormone therapy, assess metabolic conditions that increase clot risk (e.g., severe obesity, polycythemia, uncontrolled diabetes). Congenital endocrine disorders: Screen for DSDs such as androgen insensitivity, Klinefelter syndrome, or congenital adrenal hyperplasia.	Hormone therapy: Estrogen + anti-androgen for trans women; testosterone for trans men. Monitoring and dose adjustment: Regular follow-ups for ≥3 years (every 3 months in the first year, then every 6–12 months).	Patients under follow-up: Since 2004, 179 transgender patients (124 trans men, 55 trans women) have been monitored.
Gynecology	Gynecological evaluation: In trans men, uterus and ovary status is assessed via exam and pelvic ultrasound.	Gynecological pathologies: Pre-surgical evaluation includes uterine/ovarian pathologies (e.g., cysts, tumors), cervical dysplasia, or other conditions requiring treatment.	In trans men: Total hysterectomy with bilateral salpingo-oophorectomy is performed to eliminate reproductive capacity, as legally required. If lower surgery is planned, vaginectomy is also carried out.	Hysterectomy procedures: At our center, uterus and ovary removal is performed for trans men.
Urology	Physical examination: Comprehensive assessment of external genital structures. Imaging: Ultrasonography for internal genital visualization when required; pelvic CT/MRI if indicated. Hormonal and biochemical tests: Basic hormone assays (e.g., testosterone) and related labs, coordinated with endocrinology. Chromosome analysis: Karyotyping to confirm genetic sex.	Internal/external genital concordance: Evaluation for mismatches between internal organs and external genital appearance (e.g., uterus in an XY individual). Endocrine anomalies: Assessment for chromosomal or hormonal disorders (e.g., Klinefelter syndrome, androgen insensitivity). Urological health: Preoperative assessment and management of conditions such as urogenital infections, prostate disease, or testicular pathology.	In trans women (MtF): Orchiectomy and penectomy are followed by vaginoplasty, creating a neovagina from penile skin with clitoroplasty and labioplasty. In trans men (FtM): Surgery includes vaginectomy, urethroplasty, and scrotoplasty with testicular prostheses, with penile reconstruction performed either by microsurgical flap phalloplasty or metoidioplasty using the hypertrophied clitoris.	Vaginoplasty: 29 MtF patients who underwent penile inversion vaginoplasty. Falloplasty:22 trans male patients underwent this procedure.
Plastic Surgery	Preoperative evaluation: BMI, smoking status, and related factors are recorded; standard anesthesia workups (ECG, chest X-ray, blood tests) are completed before major surgeries. Before chest surgery: In trans men, breast USG/mammography is performed before mastectomy; in trans women, ≥12 months of HRT is recommended before prosthesis placement. Before phalloplasty: For microsurgical penile reconstruction, donor site vascular status (e.g., forearm arteries—Allen test) and nerve condition are assessed.	Wound-healing risks: Factors increasing postoperative complications—such as smoking, diabetes, or severe obesity—are managed (e.g., requiring smoking cessation before surgery). Realistic expectations: Patients’ desired esthetic outcomes are assessed to ensure they are achievable within realistic limits. Duration of hormone therapy: For breast surgeries, it is assessed whether patients have received adequate duration of hormone treatment.	In trans men: Mastectomy creates a masculinized chest wall, with phalloplasty (often using forearm skin) performed alongside urology when genital surgery is planned. In trans women: Breast augmentation is common, with optional facial feminization (e.g., forehead, chin, rhinoplasty). Plastic surgeons may also support vaginoplasty for flap preparation and reconstruction.	Mastectomy: One of the most frequent gender-affirming procedures at our center, performed for trans men. Breast implants: Breast augmentation is performed in trans women. Other: Phalloplasty and vaginoplasty numbers are noted above (see Urology); facial feminization surgeries are rarely performed, with only a few cases based on demand.
Otorhinolaryngology (ENT)	Voice analysis: Speech frequency, pitch range, and intensity are measured with specialized software, and objective parameters are recorded. Videolaryngoscopy: Endoscopic evaluation of the vocal cords to detect structural abnormalities (e.g., nodules, polyps, laryngitis). Speech assessment: In trans women, perceived voice femininity/masculinity is evaluated with questionnaires and standardized scales (e.g., voice quality scales) before therapy.	Laryngeal pathologies: Hoarseness-causing lesions (e.g., nodules, polyps) are treated first; conditions like severe reflux are managed before surgery. Alternative options: Before surgery, the potential for adequate vocal feminization/masculinization through training is assessed; in trans women, prolonged voice therapy is prioritized before surgical decisions.	Voice therapy: Focused on speech technique, pitch, resonance, and articulation—particularly in trans women—to align the voice with affirmed gender. Voice feminization surgery: When conservative methods are insufficient, vocal cord surgery—such as glottoplasty or the cricothyroid approach—is performed to permanently raise pitch. Adam’s apple reduction: Thyroid cartilage reduction is performed to decrease laryngeal prominence.	Voice therapy recipients: Most trans women underwent voice therapy prior to surgery; trans men were referred when necessary (e.g., for intonation training). Surgical numbers: Only a few trans women underwent voice feminization surgery at our center, with most managed conservatively. In trans men, testosterone therapy typically provided adequate voice deepening, eliminating the need for ENT surgery.

## Data Availability

No new data were created or analyzed in this study.

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
