# Peer review of "Navigating Gender-Affirming Healthcare in Adverse Political Climates: Experience from a University-Based Transgender Clinic in Türkiye"

_healthcare, 2025, doi:10.3390/healthcare13202591_

Round 1
Reviewer 1 Report
Comments and Suggestions for Authors
Many thanks for this very important, well-needed, and very interesting detailed work on what can include gender affirming care practices in a challenging socio-political context. I appreciate the comprehensive and holistic explanation of the practices adopted the Kocaeli University Hospital Gender Dysphoria Clinic. I only have a few comments.
First, I would like you to provide if possible further justifications on your choice of methodolody. There are merits in adopting an experience-based descriptive approach here, but I wonder whether in your discussion and/or introduction you could also give a short state of art on gender-affirming care practices at global level, so this can also benefit an international audience. Second, how did you clinic came to adopt/develop this approach?
Then it could be good to provide further details about the eligibility access for patients at your clinic, for instance in terms of socio-economic resources. Is care available for patients with low economic resources?
I appreciated your detailed explanation of the Turkish socio-political context but I wonder if you could expand on the current challenges faced by the clinic in regards to the recent government regulations. For instance, do the state-designated Medical Indications Assessment Board for Gender Reassignment hinder clinical decisions? Are there contradictions/challenges in this aspect? It is not very clear how much autonomy has the clinic in regard to government authority, please develop on this point.
Overall, a very informative piece of work.
Author Response
Summary
We sincerely thank Reviewer 1 for the thoughtful and constructive comments. These suggestions have substantially improved the manuscript by strengthening the methodological justification, situating the model within the global literature, and clarifying socio-economic and legal aspects of access to care. All revisions have been incorporated into the resubmitted manuscript with Track Changes. Detailed responses are provided below.
Comment 1: Please provide further justifications on your choice of methodology. There are merits in adopting an experience-based descriptive approach here, but I wonder whether in your discussion and/or introduction you could also give a short state of art on gender-affirming care practices at global level, so this can also benefit an international audience.
Response 1: We thank the reviewer for this valuable point.
- In the Abstract (page 1, line 31), we revised the Methods section to clarify that the experience-based descriptive approach was chosen specifically to capture two decades of clinical experience, institutional processes, and socio-political challenges not fully reflected in systematic reviews.
- In the Introduction (page 2, line 73), we added a new paragraph summarizing the global state of the art. This paragraph highlights the increasing emphasis on multidisciplinary, patient-centered, and rights-based models worldwide, while also pointing out persistent inequalities in middle- and low-income countries. We further emphasized that continuous psychiatric follow-up is not mandated in many countries but is gaining importance due to its role in managing comorbidities and psychosocial difficulties.
Comment 2: How did your clinic come to adopt/develop this approach?
Response 2: Thank you for raising this important point.
- In the KoUGIC Model section (page 3, line 140), after the description of the clinic’s foundation in 2004 and the first surgery in 2008, we added details on the historical development of the model.
- We clarified that plastic surgery, gynecology, and otorhinolaryngology were subsequently integrated and that the structured Gender Affirmation Board was deliberately established to ensure consensus-based decisions, distribute responsibility among clinicians, and safeguard practice within a restrictive socio-political environment.
Comment 3: Please provide further details about the eligibility access for patients at your clinic, for instance in terms of socio-economic resources. Is care available for patients with low economic resources?
Response 3: We appreciate this helpful suggestion.
- In the KoUGIC Model section (page 4, line 166), after the sentence explaining that the clinic accepts applicants at all stages, we added a paragraph on socio-economic accessibility.
- We clarified that as a public university hospital, KoUGIC provides most services under Türkiye’s national health insurance system, which allows patients with limited financial resources to access psychiatric assessments, hormone therapy, and surgical interventions without direct costs.
- At the same time, we acknowledged that indirect expenses such as travel, accommodation, or medications not fully covered by insurance remain barriers for patients from disadvantaged or rural areas.
Comment 4: Please expand on the current challenges faced by the clinic in regard to the recent government regulations. For instance, do the state-designated Medical Indications Assessment Board for Gender Reassignment hinder clinical decisions? Are there contradictions/challenges in this aspect? It is not very clear how much autonomy has the clinic in regard to government authority, please develop on this point.
Response 4: We thank the reviewer for highlighting this essential issue.
- In the KoUGIC Model section (page 4, line 193), after the sentence describing the centralization of decision-making under the state-designated Medical Indications Assessment Board, we clarified the respective roles of the clinic, the Board, and the court.
- We explained that the report produced by KoUGIC’s multidisciplinary board is considered a medical opinion rather than a formal authorization. The final legal decision regarding eligibility for surgery is made by the court.
- In practice, this means that even if our board deems a patient suitable for surgery, the court may deny permission. We highlighted that this dual structure reduces clinical autonomy and can cause delays and inconsistencies between medical expertise and legal authority.
Overall Response:
We sincerely thank Reviewer 1 for the constructive and detailed comments. In response, we revised the manuscript in four areas:
- Strengthened the methodological justification (Abstract, page 1, line 31).
- Added global context and state-of-the-art paragraph (Introduction, page 2, line 73).
- Expanded on the historical development of the KoUGIC model and socio-economic accessibility (KoUGIC Model, page 3, line 140; page 4, line 166).
- Clarified the impact of recent government regulations, the role of the state-designated Board, and the limits of clinical autonomy (KoUGIC Model, page 4, line 193).
We believe these revisions have improved the clarity, scope, and international relevance of the manuscript.
Reviewer 2 Report
Comments and Suggestions for Authors
This article offers a really interesting insight into the current healthcare system in Türkiye and I think it is really useful for other clinicians in this area to read.
Line 84 - It is probably better to call it “gender affirmation treatments” rather than “gender transition”.
Line 121 - If you know when these other services were added it would be great to know the years, although this is just for interest so doesn’t matter if you don’t have them.
The only big question I have about this article is that it seems to be just descriptive and does not have any actual research in it. If this is what healthcare are happy to print then that would be fine. I am probably just more used to papers that describe a study or a systematic review and this feels more like a chapter in a book.
Also, there is no mention of non-binary people or the treatments they require. Would be interesting to see this.
Author Response
Summary
We sincerely thank Reviewer 2 for the encouraging feedback and constructive comments. The suggestions have allowed us to improve terminology, add contextual details, and address important issues such as non-binary individuals. All revisions are incorporated in the resubmitted manuscript with Track Changes. Detailed responses are provided below.
Comment 1 (Line 84): It is probably better to call it “gender affirmation treatments” rather than “gender transition.”
Response 1: We thank the reviewer for this helpful suggestion. We have replaced the term “gender transition” with “gender affirmation treatments” throughout the manuscript to ensure clarity and alignment with current terminology.
- Revisions made:Page 2, line 63 and all relevant occurrences across the text.
Comment 2 (Line 121): If you know when these other services were added it would be great to know the years, although this is just for interest so doesn’t matter if you don’t have them.
Response 2: Thank you for pointing this out. We have added the available years of integration for different specialties in the KoUGIC model. Psychiatry and endocrinology initiated the program in 2004, urology performed the first surgery in 2008, and plastic surgery, gynecology, and otorhinolaryngology were subsequently incorporated in the following years.
- Revisions made:Page 3, line 140 (KoUGIC Model section).
Comment 3: The article seems to be just descriptive and does not have any actual research in it.
Response 3: We appreciate this observation. We would like to clarify that the article was intentionally designed as an experience-based practice report and therefore was submitted to the journal as a protocol rather than an original research article. Our aim was not to present new empirical data but to share nearly two decades of accumulated clinical practice, institutional processes, and socio-political challenges of gender-affirming care in Türkiye. Given the unique and difficult context of working in a country where conducting such studies can be challenging, our primary goal was to demonstrate how multidisciplinary gender-affirming care can be sustained in Türkiye and to provide an example that may be useful beyond Western cultural settings. We clarified this in the Abstract (Methods section, page 2, line 31) and in the Introduction to avoid misunderstanding.
Comment 4: There is no mention of non-binary people or the treatments they require.
Response 4: We thank the reviewer for highlighting this important point. We have now added a paragraph addressing the situation of non-binary individuals in Türkiye.
- We noted that individuals with gender dysphoria in Türkiye often hesitate to seek healthcare even for daily health problems due to stigma and discrimination. Those who come to university hospitals usually request gender-affirming medical care (hormones or surgery). Since non-binary people generally do not request such treatments, they rarely present to hospitals.
- Being non-binary is even more challenging in Türkiye, as it is often less socially recognized than binary transgender identities. Sometimes, due to social pressure and difficulty of acceptance, non-binary individuals initially feel compelled to request binary-oriented treatments. During group psychotherapy at KoUGIC, however, some individuals realize and accept their non-binary identity, and in such cases, they often withdraw from hormone or surgical requests while continuing to receive individual psychotherapy support if needed.
- Revisions made:KoUGIC Model section, Page 4, after line 173.
Overall Response:
We sincerely thank Reviewer 2 for these insightful comments. In response, we:
- Replaced terminology with “gender affirmation treatments” (line 63, and throughout manuscript).
- Added available years of integration for other specialties (line 140).
- Clarified the descriptive, protocol/practice-report nature of the article (Abstract, page 2, line 31; Introduction), emphasizing that it was intentionally submitted as a protocol rather than an original research article.
- Added a dedicated explanation regarding non-binary individuals, their challenges, and the type of support available at KoUGIC (page 4, after line 173).
These revisions have improved the clarity, inclusivity, and contextual depth of the manuscript.
Reviewer 3 Report
Comments and Suggestions for Authors
The authors have written a well-structured and informative article.
Below are some suggestions to improve it.
The text is largely descriptive and does not provide detailed statistical evidence on the effectiveness of psychotherapy, surgical success, or patient satisfaction.
It also fails to provide a critique of the limitations or practical challenges associated with training and practice. For surgeries and hormonal procedures, only the overall number of cases is listed, and more detailed information, such as follow-up outcomes, complications, or successes, is not provided.
Repetition of information in different sections, such as some surgeries and care in the urology and plastic surgery departments, could be summarized. Overall, the text repeats a lot of information. Similar concepts (e.g., family role, minority stress, psychotherapy) are repeated several times in different sections, making the text lengthy and somewhat incoherent.
Author Response
Summary
We sincerely thank Reviewer 3 for the positive evaluation and constructive feedback. These comments have allowed us to strengthen the manuscript, especially by clarifying its descriptive nature, expanding on limitations, and improving coherence. All revisions have been incorporated into the resubmitted manuscript with Track Changes.
Comment 1: The text is largely descriptive and does not provide detailed statistical evidence on the effectiveness of psychotherapy, surgical success, or patient satisfaction.
Response 1: We agree with the reviewer. This article was designed as a practice-based descriptive report rather than an empirical study, in order to document two decades of clinical experience and institutional processes at KoUGIC. We clarified this in the Abstract – Methods (page 2, line 31) and reiterated it in the Discussion (page 11, line 394). In addition, in the Limitations section (page 12, line 449), we explicitly noted that statistical outcome data on psychotherapy, surgical procedures, and patient satisfaction are not systematically presented in this paper, and emphasized that future research should address these outcomes.
Comment 2: The manuscript does not provide a critique of the limitations or practical challenges associated with training and practice.
Response 2: Thank you for pointing this out. In the Limitations section (page 12, line 449), we added a critical discussion of training and practice challenges. Specifically, we emphasized the shortage of specialists trained in transgender health, insufficient curricular content on LGBT+ health in medical education, and the persistence of stigma and provider bias among healthcare professionals in Türkiye. We would also like to note that we actively recognize these gaps and have taken steps to address them at KoUGIC. For example, we regularly include medical students and interns in our Gender Affirmation Board meetings and clinical processes, fostering hands-on experience in transgender health. Thus, while these training deficiencies represent a systemic limitation at the national level, our clinic has developed internal strategies to mitigate them and contribute to improving education and practice in this field.
Comment 3: For surgeries and hormonal procedures, only the overall number of cases is listed, and more detailed information, such as follow-up outcomes, complications, or successes, is not provided.
Response 3: We appreciate this suggestion. We clarified in the Limitations section (page 12, line 453) that although Table 2 provides the overall number of cases, standardized follow-up data on surgical outcomes, complications, and long-term success rates are limited. We highlighted this as an important direction for future systematic studies. This is also the reason why we submitted the article as a protocol/practice report rather than an original research article. Our primary aim was to present a positive, long-lasting example of a gender clinic model outside of Western cultural settings, demonstrating how multidisciplinary gender-affirming care can be implemented and sustained in Türkiye’s challenging sociopolitical context.
Comment 4: There is repetition of information in different sections (e.g., some surgeries in urology and plastic surgery are described more than once; family role, minority stress, psychotherapy are repeated).
Response 4: We thank the reviewer for this observation. In the revised manuscript, we carefully reviewed the text and reduced repetition to improve flow and coherence. Specifically, descriptions of surgeries in Urology and Plastic Surgerysections were consolidated, and overlapping discussions of family role, minority stress, and psychotherapy were streamlined. Repetitive parts of the Discussion were also shortened. These changes made the text more concise and coherent.
Overall Response:
We sincerely thank Reviewer 3 for these helpful suggestions. In response, we:
- Clarified the descriptive nature of the article (Abstract, page 2, line 31; Discussion, page 11, line 394).
- Expanded the critique of limitations and training/practice challenges (Limitations, page 12, line 449).
- Clarified the absence of standardized outcome data for surgical and hormonal interventions (Limitations, page 12, line 453).
- Reduced repetition across sections, including Urology, Plastic Surgery, and Discussion (pages 11-12).
We believe these revisions have substantially strengthened the manuscript.